# Axon Guidance Molecules and Pain

**DOI:** 10.3390/cells11193143

**Published:** 2022-10-06

**Authors:** Elisa Damo, Manuela Simonetti

**Affiliations:** Institute of Pharmacology, Medical Faculty Heidelberg, Heidelberg University, Im Neuenheimer Feld 366, 69120 Heidelberg, Germany

**Keywords:** chronic pain, Wnt signaling, ephrin, Eph receptors, semaphorins, plexins, neuronal plasticity

## Abstract

Chronic pain is a debilitating condition that influences the social, economic, and psychological aspects of patients’ lives. Hence, the need for better treatment is drawing extensive interest from the research community. Developmental molecules such as Wnt, ephrins, and semaphorins are acknowledged as central players in the proper growth of a biological system. Their receptors and ligands are expressed in a wide variety in both neurons and glial cells, which are implicated in pain development, maintenance, and resolution. Thereby, it is not surprising that the impairment of those pathways affects the activities and functions of the entire cell. Evidence indicates aberrant activation of their pathways in the nervous system in rodent models of chronic pain. In those conditions, Wnt, ephrin, and semaphorin signaling participate in enhancing neuronal excitability, peripheral sensitization, synaptic plasticity, and the production and release of inflammatory cytokines. This review summarizes the current knowledge on three main developmental pathways and their mechanisms linked with the pathogenesis and progression of pain, considering their impacts on neuronal and glial cells in experimental animal models. Elucidations of the downstream pathways may provide a new mechanism for the involvement of Wnt, ephrin, and semaphorin pathways in pain chronicity.

## 1. Introduction

Pain is an essential physiological sensory and emotional experience that is necessary for protecting the integrity of the body. Under certain circumstances, physiological pain undergoes maladaptive processes and becomes chronic, i.e., a pain that persists for more than three months. Chronic pain affects around one-fifth of the world’s population, but for a considerable percentage of patients, the current therapies and conventional analgesics are not successful. Better knowledge of the pathophysiological mechanisms involved in the development and maintenance of chronic pain is needed to achieve more specific and efficient therapies.

Different classes of molecules are well known to mediate pain. At the periphery, nerve injury, inflammatory condition, or cancer growth can recruit different kinds of cells, such as Schwann cells, fibroblast, dendritic cells, epithelial cells, mast cells, lymphocytes, macrophages, and neutrophils. These cells secrete primary mediators such as chemokines, cytokines, neuropeptides, and growth factors, generating an “inflammatory soup.” Primary mediators directly activate or sensitize sensory nerve endings by changing ion channel properties, altering gene expression, or inducing post-transcriptional modification (among other means). These changes result in increased excitability, spontaneous activity, and the release of secondary mediators at the spinal level.

Secondary mediators such as colony stimulating factor 1 (CSF-1) and chemokine (C-C motif) ligand 21 (CCL21) activate spinal cells such as astrocytes and microglial cells, causing them to release tertiary mediators, such as brain derived neurotrophic factor (BDNF), tumor necrosis factor alpha (TNF-α), and interleukin-1β (IL-1β), that increase excitatory transmission and attenuate the inhibitory synaptic transmission of the spinal dorsal horn (SDH) neurons. More recently, secreted extracellular vesicles or exosomes containing microRNAs have also been shown to be involved.

Often, in adulthood, in certain physiological and pathological conditions, there is a re-awakening of genes and proteins that sculpt the developing nervous system. Over the past 20 years, molecules that are crucial during embryogenesis and development processes, such as the wingless-related integration site (Wnt) morphogens, and the axon guidance molecules, semaphorins and ephrins, have emerged as important factors involved in the pathophysiology of various forms of chronic pain. Indeed, evidence indicates aberrant activation of their pathways in the nervous system in rodent models of chronic pain.

Wnt ligands (Wnts) are a large family of secreted glycoproteins whose signaling pathway is highly conserved and plays a key role in intercellular interaction and communication. Wnts are strongly involved in cell differentiation, migration, and proliferation and in the development of the central and peripheral nervous system (CNS and PNS). Furthermore, neuronal Wnt signaling participates in various postnatal processes, such as brain plasticity and synaptic physiology. In the adult mouse brain, Wnts can affect memory formation and the development of activity-dependent long-term potentiation (LTP), a form of persistent increase in synaptic strength driven by neuronal activity. Moreover, a variety of neurologic disorders, including psychiatric and neurodegenerative diseases, such as Alzheimer’s disease, Parkinson’s disease, schizophrenia, and chronic pain, have been associated with the dysregulation of Wnt signaling [1,2,3,4].

Ephrin receptors (Eph receptors) are a large family of receptor tyrosine kinases (RTK) involved in embryonic development that bind the membrane-bound proteins called ephrins (ligands) [5]. Ephrins regulate the development of many organs and tissues [6,7], including the CNS, where they mediate axon repulsion [8]. The Eph–ephrin system regulates adult tissue homeostasis and tumor development [7]. Furthermore, in the adult CNS, ephrins are expressed by neurons and glial cells and mediate synaptogenesis and synaptic plasticity [9,10].

Semaphorins are an important family of conserved molecules that are crucial for driving axons to their targets during the development of the nervous systems [11]. They can be soluble or membrane-associated via a transmembrane segment or via a glycosylphosphatidylinositol (GPI) tail and binds the transmembrane proteins plexins and neuropilins [12]. The correct semaphorin–plexins signaling during development is fundamental for the formation and organization of neuronal circuitry. Indeed, its dysregulation has been linked to developmental diseases of the nervous system such as autism and schizophrenia, among others [13,14,15,16], and to neurodegenerative diseases [17].

These three signaling systems share many common elements: they are redundant, interact with each other, and are active and necessary for the processes of embryonic development and the proper functioning of the CNS in the adult. Indeed, alterations in each one of these signaling pathways lead to neurodegenerative diseases and the development of certain tumors. Interestingly, their involvement has also been recently described in chronic pain.

In the adult CNS, members of these important protein families are expressed by different cell types involved in pain, including neurons, astrocytes, and microglia.

Under physiological conditions, astrocytes exert a wide range of functions in the CNS. Among others, they provide metabolic support, regulate synaptic plasticity, control blood flow and iron homeostasis, neutralize reactive oxidative substances (ROS), and maintain the structure and function of the blood–brain barrier. Astrocytes react to any perturbation of CNS homeostasis, developing a complex response whose output can be beneficial or deleterious, depending on the type of injury, the metabolic state, and the crosstalk with microglia and neurons (reviewed in [18,19]).

Microglia are macrophage-like cells of the CNS which regulate tissue maintenance. Microglial cells constantly monitor the surrounding environment to react promptly to any element that disturbs the homeostasis of the CNS. They are extremely plastic and capable of rapidly changing their phenotypes in response to external conditions and stimuli that are detected due to the wide variety of receptors they express. Numerous pieces of evidence demonstrate the involvement of astrocytes and microglia in the development of chronic pain, both at the spinal level and at the brain level [20,21,22,23].

This review summarizes recent advances in finding determinant molecules in chronic pain, with a particular focus on the Wnt, Eph–ephrin, and semaphorin–plexin signaling.

## 2. Wnt Signaling in Pain

The heterogeneity of Wnt signaling starts with the ligand itself. There are 19 members of the Wnt family in humans and rodents, each one with a different expression pattern and function. These ligands bind different kinds of receptors: the classical Frizzled (Fzd) receptors (a family of G protein-coupled receptors that comprises 10 members in vertebrates), which are frequently associated with co-receptors, such as low-density lipoprotein receptor-related protein 5/6 (LRP5/6), the RTKs, receptor-like tyrosine kinase (Ryk), receptor tyrosine kinase-like orphan receptor 2 (Ror2), protein-tyrosine kinase-7 (PKT7), and muscle-specific kinase (MuSK), or proteoglycans. Usually, many ligands can bind the same receptor and one ligand can bind different receptors, increasing the complexity of Wnt signaling.

Depending on the cell type and the cellular metabolic state, Wnt ligands can engage different pathways. The most common are the canonical or β-catenin-dependent pathway and two non-canonical pathways, the planar cell polarity (PCP) pathway and the calcium (Ca^2+^) pathway. Briefly, the canonical pathway is mediated by β-catenin stabilization and its nuclear translocation, resulting in β-catenin-dependent gene transcription. The PCP pathway involves the activation of the small GPTases Rho and Ras-related C3 substrate botulinum toxin 1 (Rac-1), which activate Rho-associated protein kinase (ROCK), c-Jun amino (N)-terminal kinase (JNK), mediating cytoskeletal rearrangement and gene transcription. Fzd-dependent activation of phospholipase C (PLC) mediates the Wnt-Ca^2+^ pathway in an inositol 3-phosphate (IP3)-dependent manner. This increases intracellular Ca^2+^ transients and leads to the activation of several Ca^2+^-dependent kinases that promote gene transcription and phosphorylation of signaling proteins and ion channels. Due to its crucial functions, Wnt signaling is tightly regulated at several levels (reviewed in [24]). Synaptic-activity-regulated Wnt signaling has been shown to be critical for the functional and structural remodeling of synapses [25].

### 2.1. Involvement of Wnt Signaling in Chronic Pain

In the last decade, a growing body of evidence has demonstrated the involvement of the Wnt signaling pathway in the context of chronic pain, both in patients and in several pre-clinical mouse models of pain [26].

Wnt signaling is activated at different levels along the pain pathway and in diverse cell types, depending on the chronic pain condition or the pain model studied (Figure 1). Most studies, using preclinical pain models, focused on Wnt pathway activation in dorsal root ganglia (DRGs) and the spinal cord. In contrast, reports considering the supraspinal level are almost completely absent.

### 2.2. Neuronal Wnt Signaling Dysregulation in Chronic Pain

Peripheral sensitization is a mechanism underlying the development of chronic pain. Generally, it originates from small molecules released by different cell types in pathological conditions that can activate and modulate the nociceptors. These small molecules can alter the properties of nociceptive ion channels such as transient receptor potential cation channel subfamily V member 1 (TRPV1), transient receptor potential ankyrin 1 (TRPA1), voltage-dependent sodium channel (Na_v_) 1.7, and Na_v_1.8 via post-translational modifications such as phosphorylation, leading to increased neuronal activity, reduced activation thresholds, or increased currents [27].

Our laboratory and others demonstrated that Wnt ligands sensitize peripheral nociceptors [28,29]. In a model of cancer pain (osteolytic fibrosarcoma cells in the calcaneus bone), we found that non-neuronal cells release Wnt3a, which triggers DRG neurons via non-canonical pathways enhancing the membrane translocation of P2X purinoceptor 3 (P2X3) and TRPV1 receptors [28]. Moreover, blocking Wnt-Fzd3 signaling in sensory neurons is sufficient to attenuate cancer pain sensitization.

More recently, He et al. have shown that Wnt5b-Ryk signaling is involved in bone cancer pain via Ca^2+^/calmodulin-dependent protein kinase II (CaMKII)-dependent, P2X3-mediated increased DRG excitability [30]. In a mouse model of diabetic neuropathic pain (DNP), Wnt5a is released from A-fiber in a Wnt ligand secretion mediator (GPR177)-dependent way and directly binds and activates TRPV1 receptors expressed by the neighboring C-fibers [31]. Interestingly, in two different rat models of neuropathic pain, paclitaxel-induced and streptozotocin (STZ)-induced pain, Wnt signaling pathway inhibitors NSC668036 and PNU74654 reverse the reduction in intraepidermal nerve fiber density (IENFD) [32,33], used as a clinical marker of chemotherapy-induced peripheral neuropathy [34] and diabetic peripheral neuropathy [35]. These findings underline the importance of Wnt signaling in mediating neuropathic pain also at the level of peripheral terminals, although the study of this phenomenon is only beginning.

In cancer conditions and different models of chronic pain, such as tumor-cell-induced pain (TCI), chemotherapy-induced neuropathic pain (PTX-induced pain), and chronic constriction injury (CCI, a model of neuropathic pain with a strong inflammatory component), Wnt ligands such as Wnt3a, Wnt5b, and Wnt10a; Wnt receptors such as Fzd8 and Ryk; and signaling molecules such as β-catenin, GSK-3β, and TCF4 are upregulated in DRGs [29,36].

The increased expression of Wnt ligands in DRG supports their release from the sensory afferents into the spinal cord in an activity-dependent manner driven by pain stimuli [29,37]. Indeed, neuronal activity controls both the expression and the secretion of Wnts (reviewed in [38]).

At the spinal cord level, Wnt signaling can modulate the pain sensation by acting directly on neurons and regulating synaptic plasticity, or by recruiting non-neuronal cells such as microglia and astrocytes. Wnt ligands are known to be important modulators of synaptic plasticity [39], a phenomenon that is well established as a crucial mechanism underlying chronic pain [40,41]. Activation of Wnt signaling via NMDA-receptor-mediated synaptic Wnt3a release induces LTP, a form of synaptic plasticity [25]. Moreover, the inhibition of Wnt signaling blocks LTP. Accordingly, Wnt family members are upregulated into the spinal cord in several chronic pain conditions and mouse pain models.

Activation of the Wnt/β-catenin canonical pathway leads to the increased production and secretion of pro-inflammatory cytokines and BDNF, which enhance neuronal excitability and synaptic plasticity [28,41,42]. Furthermore, the canonical pathway regulates the N-methyl-D-aspartate receptor subunit 2B (NR2B)- and Ca^2+^-dependent signals in the dorsal horn [29]. In the CCI and spinal nerve ligation (SNL) neuropathic pain models, Wnt5b/Ryk signaling contributes to the development of neuropathic pain; these proteins are upregulated in DRGs and the spinal cord after nerve injury [42]. Ror2 plays a relevant role in CCI-induced neuropathic pain: modulating synaptic plasticity via phosphorylation of NR2B, protein kinase C (PKC), and Src family kinases in the spinal cord [43]. Furthermore, in models of nerve injury or inflammatory pain, Wnt5a is secreted in an activity-dependent manner and mediates chronic pain via modulation of synaptic spines [37]. Importantly, blocking Wnt pathways is sufficient to reduce neuropathic pain. Indeed, blocking Ryk signaling decreases neuronal excitability, lowers the enhanced synaptic plasticity between C-fibers and dorsal horn neurons, the nerve-injury-induced increased intracellular Ca^2+^, and activation of the NR2B receptor [42,44]. Moreover, inhibiting Wnt3a/β-catenin signaling with the Wnt inhibitor IWP-2 reduced CCI-induced neuropathic pain, inhibiting synaptic plasticity in the spinal cord [45]. These results highlight the contributions of neuronal Wnt signaling to the CCI-induced neuropathic pain via both canonical and non-canonical pathways.

Neuronal Wnt signaling in the spinal cord has an important role in HIV-induced neuralgia. Several Wnt ligands and β-catenin are upregulated in the spinal cord of HIV patients that experience chronic pain, but not in the pain-free ones [46]. The injection of the viral protein HIV1-gp120, a model of neuropathic pain associated with HIV infection, induces Wnt3a upregulation in microglia [47,48], Wnt5a, and pro-inflammatory molecules IL-1β, IL-6, and TNF-α at the spinal cord level; intrathecal injection of Wnt5a antagonist Box5 significantly reduces the levels of inflammatory cytokines [46]. Furthermore, the recombinant protein gp120 activates neurons by directly stimulating their NMDARs [49,50], leading to the synthesis and secretion of Wnt5a [51]. Indeed, NMDAR is a key mediator of Wnt5a [52], which plays a critical role in the differentiation and plasticity of excitatory synapses [53,54]. Recently, it was shown that Wnt5a could mediate HIV-related pain also via the Ror2/MMP2/IL-1β pathway [55].

In addition, to directly modulate neuronal excitability and synaptic plasticity, Wnt signaling can induce neuroinflammation and recruit glial cells (see below).

### 2.3. Wnt Signaling Pathway in Glial Cells in Neuropathic Pain

Neuron–glia crosstalk through Wnt signaling may play a key role in the pathogenesis of several diseases, such as neurodegenerative conditions and chronic pain. Glial cells respond to secreted Wnt ligands which induce pro-inflammatory activation of glial cells, characterized by morphological changes and the release of pro-inflammatory mediators. The glial response, in turn, modulates neuronal function and the plasticity of neural circuits [38]. Interestingly, regardless of the type of tissue involved or the kind of injury, the endogenous β-catenin-dependent Wnt signaling pathway is frequently activated at the site of tissue damage [56] or along the pain pathway. Despite the experimental evidence is pointing out a significant role of the Wnt family in the physiological and pathological functioning of the spinal cord, cell-type-specific information is still lacking [57,58].

#### 2.3.1. Astrocytes

Emerging evidence supports the role of canonical and non-canonical Wnt pathways as activators of astrocytes [59,60]. Under physiological conditions, astrocytes express a large panel of Wnt-related proteins. Astroglia are assumed to be the main source of Wnt ligands in the spinal cord, and harboring a wide variety of Wnt receptors, they are considered the main actor in the multidirectional astrocyte-neuron-microglia crosstalk [61,62]. Depending on the kind of injury and on the metabolic state of astroglia when activated, Wnt pathways modulate cell proliferation, glutamate uptake, the expression of glutamate transporters, pro-inflammatory cytokines, trophic factors, potassium, and water channels [44,62,63,64,65,66,67].

Under physiological conditions, Wnt receptors show cell- and spatial-specific expression patterns [68] that are altered after injury, indicating different cell-specific physiological roles at the spinal level [58]. For example, after CCI, Fzd1 is transiently upregulated in spinal neurons, whereas Fzd8 is persistently upregulated in spinal astrocytes and satellite cells in DRGs [29]. Furthermore, in the SNL model, Ryk is overexpressed on unmyelinated fibers, promoting the production and release of chemokine (C-C motif) ligand 2 (CCL2), which activates microglia [44]. CCI-induced nociceptive hypersensitivity is significantly attenuated by hyperbaric oxygen treatment via suppressing the spinal kindlin-1/Wnt10a signaling pathway and activation of astrocytes [69]. Kindlin-1 is a β-integrin binding protein that participates in the induction of inflammation and pain sensitization [70,71]. In CCI-treated rats, kindlin-1 is shown to be upregulated in spinal astrocytes [70]. In accordance, downregulation of kindlin-1 reduces mechanical allodynia and astrocytic activation [72]. This effect could be mediated by the modulation of Wnt expression by kindlin-1, as demonstrated in keratinocytes [73]. Interestingly, the analgesic effect of dexmedetomidine, an agonist of α2-adrenergic receptors, used to treat a refractory form of neuropathic pain, administrated at late time points in the STZ-induced diabetic neuropathic pain model, is mediated by inhibition of the Wnt10a/β-catenin signaling pathway and astrocytic activation [74]. Moreover, early-time-point dexmedetomidine administration relieves mechanical and thermal hyperalgesia by impeding microglial activation [75].

In the CCI model, Ryk is upregulated in astrocytes and microglia in the SDH, and in satellite cells in DRGs [42]. Wnt5a, Ryk, and ROR2 are overexpressed in different pain models, such as SNL, hind-paw injection of capsaicin, and HIV1-gp120 intrathecal injection [76]. Importantly, in a rat neuropathic model of chronic post-thoracotomy pain (CPTP) and in other mouse models of neuropathic and inflammatory pain, the specific Wnt5a antagonist Box5 considerably inhibits the activation of astrocytes in the spinal cord and relieves mechanical allodynia and thermal hyperalgesia [37,77,78,79]. In the CPTP model, Liu and colleagues found that Ror2 predominantly co-localizes with astrocytes and modulates their activation, leading to a pro-inflammatory phenotype named A1 [55]. Indeed, the knockdown of Ror2 promoted the neuroprotective phenotype of astrocytes (A2) versus the toxic one (A1) and attenuates mechanical hyperalgesia and thermal allodynia. Interestingly, Ror2 downregulation reduces the expression of C3aR in spinal astrocytes, suggesting that the modulatory effect of Ror2 on astrocytes phenotype can be mediated via C3aR expression [55]. Wnt5a is upregulated only in pain-positive HIV1 patients [46], whereas recombinant HIV1-gp120 induces Wnt5a neuronal release in an activity-dependent manner, causing hyperactivation of neurons [80] and astrocytes [81]. The HIV1-gp120-induced astrogliosis is sustained by the neuron to astrocyte Wnt5a-Ror2 signaling, and it is essential for HIV-associated pain sensitization [81]. Both neuronal Wnt5a knockdown and astrocytic Ror2 knockdown abolish HIV1-gp120-induced astrogliosis and mechanical hyperalgesia [81].

Taken together, these results indicate that activation of Wnt signaling contributes to the activation of astrocytes in the spinal cord, leading to neuroinflammation and chronic pain.

#### 2.3.2. Microglia

Primary microglial cells and microglia-like cell lines respond to recombinant Wnt3a or Wnt5a application thanks to the localization on their membranes of a variety of Wnt receptors. Wnt3a and Wnt5a induce increased synthesis of pro-inflammatory molecules such as cytokines, chemokines, and cyclooxygenase 2 (COX2), and exacerbates the release of *de novo* synthesized IL-6, IL-12, and TNFα [82,83], leading to neuroinflammation [84]. Interestingly, when applied to cultured lipopolysaccharide (LPS)-primed microglial cells, recombinant Wnt3a and Wnt5a prompt dose-dependent downregulation of IL-6, COX-2, and TNFα expression, supporting a dual role of microglia as a pro or anti-inflammatory player, depending on the surrounding environment [85,86]. Furthermore, Wnt3a applied on primary microglial cells can induce exosome secretion, without inducing a neurotoxic pro-inflammatory phenotype [87], underlining the major plasticity of microglia in responding to Wnt ligands.

Although Wnt pathways can prevent microglial activation and alleviate neuroinflammation [84,88,89,90], most studies indicate the involvement of Wnt pathways in the polarization of microglia toward a pro-inflammatory phenotype [86,91,92,93].

In several pain models in rodents, Wnt pathways induction activates microglial cells through different molecules: fractalkine (FKN) and BDNF, among others. In the HIV-1 gp120-induced pain model, activated microglia mediate synaptic degeneration. Interestingly, HIV infection prompts an increase in FKN [94,95,96], a neuronal protein that regulates microglia-dependent synaptic phagocytosis. Since FKN is mainly expressed by neurons, and its CX3C chemokine receptor 1 (CX3CR1) is specifically present in microglia, the FKN pathway establishes signaling between neurons and microglia that leads to the regulation of synaptic pruning [97,98]. Recently, it has been shown that the HIV1-gp120 protein leads to the upregulation and release of Wnt3a in an NMDAR activity-dependent manner, resulting in activation of the β-catenin pathway and induction of FKN transcription in neurons [99], ultimately resulting in synaptic degeneration of the neural spinal pain circuit. Furthermore, NMDAR antagonist DL-2-amino-5-phosphonovaleric acid (APV), the endogenous Wnt antagonist dickkopf-related protein 1 (DKK1), and knockout of CX3CR1, alleviate HIV1-gp120-induced mechanical allodynia in mice, suggesting a critical contribution of the Wnt/β-catenin/FKN/CX3CR1 pathway to HIV1-gp120-induced pain [99]. Moreover, HIV1-gp120-induced neuropathic pain is mediated by the microglial release of BDNF, a crucial neuromodulator of pain transmission [100]. Indeed, Wnt inhibitors block HIV1-gp120-induced BDNF release and subsequent induction of chronic pain, supporting a strong contribution of the Wnt pathway to spinal microglia activation, BDNF release, and chronic pain [101,102]. In a chemotherapy-induced neuropathic pain model, DKK1 significantly reduces capsaicin-induced inflammatory pain by blocking BDNF release from microglia, whereas the tankyrase inhibitor IWR-1-endo attenuates mechanical hyperalgesia [103], inhibiting the activation of astrocytes, microglia, and TNF-α, and CCL2 and MAPK/ERK signaling in the spinal cord [103].

Blocking Wnt signaling shows amelioration of neuropathic pain in other rodent pain models too. In fact, Zhang et al. showed that intrathecal injection of a Wnt signaling inhibitor, IWP-2, strongly diminishes both mechanical and thermal sensitization in CCI-operated rats via suppressing microglial reaction in the spinal cord [29]. Meanwhile, targeting the Wnt/β-catenin signaling pathway with a tankyrase inhibitor XAV-939 suppresses the activation of microglia in the spinal cord and alleviates mechanical hypersensitivity in rats that undergo partial sciatic nerve ligation (pSNL) [104]. The inhibition of β-catenin-independent Wnt pathways has recently been shown to reduce chronic pain via acting on microglia. For example, in a model of adjuvant-induced arthritis (AIA), the flavonoid crocin alleviates neuropathic pain by targeting Wnt5a signaling and microglia activation [105]. Interestingly, astrocytes in the adult mouse brain express high levels of Wnt5a, which could serve as a novel astroglia–microglia communication pathway to be targeted in chronic pain conditions.

Recently, it was shown that the activation of the receptor complex DAP12-TREM2 contributes to the development of neuropathic pain. DAP12 signaling is triggered after nerve injury, whereas the direct activation of TREM2 induces mechanical allodynia in naïve mice. Moreover, DAP12-deficient mice fail to develop allodynia after nerve injury [106]. DAP12 forms a receptor complex with TREM2 on the microglial membrane. Activation of this complex is associated with many physiological functions of microglia, and pathological conditions such as neurodegenerative diseases or chronic pain (summarized in [107,108]). Furthermore, the DAP12-TREM2 complex is involved in the survival of microglial cells via activation of Wnt/β-catenin signaling [106,109,110].

Despite both Wnt signaling and microgliosis mediating maladaptive processes such as chronic pain, they are necessary for positive effects such as adult neurogenesis and synaptic plasticity [111]. Thus, strict control of the balance between activation and inhibition of these phenomena is necessary for the maintenance of tissue homeostasis and proper physiological functions of the body. Interestingly, a recent paper demonstrated the need for early inflammation to reduce the risk of developing chronic pain later. The use of steroids or non-steroidal anti-inflammatory drugs (NSAIDs) and neutrophil depletion delayed the resolution of pain in animal models [112]. Therefore, correct balance and timing between activation and inhibition of certain cell types are necessary for a positive physiological outcome.

## 3. Eph–Ephrin Signaling in Chronic Pain

Eph receptors constitute the largest family of RTKs. So far, 14 Eph receptors have been identified in mammals. They are all transmembrane proteins and have been divided into two classes according to the similarity of their extracellular domains. Whereas the intracellular domains are highly conserved, the differences at the extracellular level determine the affinity to different membrane-associated ligands, the ephrins. Ephrins are divided into two major classes according to the way they bind to the cytoplasmic membrane: class A (ephrinAs) ones associate with the membrane via a GPI-tail and bind promiscuously with class A Eph receptors (EphAs), and class B (ephrinBs) ones have a transmembrane segment and a cytoplasmic tail and bind promiscuously the class B receptors (EphBs). Exceptions are the EphA4 and EphB6 receptors which bind ephrins of both classes. Whereas the not-activated ephrins are localized and concentrated in membrane microdomains called lipid rafts [8], the Eph receptors are distributed across the cell membrane. Following the binding to ephrin, the Eph receptor activates its kinase domain and undergoes auto-phosphorylation, forming a hetero-tetramer consisting of two ligands and two receptors. More tetramers can cluster, forming bigger signaling complexes [8,113,114]. The degree of clustering determines specific outcomes by regulating differential downstream pathways [7,115].

The Eph–ephrin system controls many cellular processes that depend on rapid changes in morphology or mobility. Indeed, when this system is activated, it mediates alterations in the cytoskeleton and the phenomena of cell attraction and repulsion [116]. Eph–ephrin binding triggers a bidirectional signal. In the forward signaling, the signal is activated in the cell expressing the receptor and depends on the auto-phosphorylation of the cytoplasmic kinase domain of the Eph receptor. The signal activated in the ligand-expressing cell is called reverse signaling. The ephrinB-mediated reverse signal depends on phosphorylation of the cytoplasmic domain mediated by src kinases. In the case of ephrinA, the reverse signaling seems to involve other transmembrane proteins, such as the low-affinity receptor for nerve growth factor p75. While the forward signal mediates mostly repulsion, the reverse signal mediates both repulsion and attraction, or adhesion, depending on the affinity of receptor–ligand binding. In addition to *in trans* (between two different cells), the interaction between Eph receptor and ephrin can also occur *in cis* (on the same cell). In general, *cis* interactions are inhibitory concerning signal activation [117].

Furthermore, the Eph–ephrin system recruits and cross-activates other signaling pathways in an intricate network capable of governing complex biological responses and processes. The final output depends on the cellular context in which the signaling is activated (reviewed in [118]). The Eph–ephrin system can interact physically or through signaling molecules with other membrane receptors, such as fibroblast growth factor receptors (FGFR), Ryk, and the cytokine receptor CXCR4. This system interacts with adhesion molecules such as integrins [119,120,121,122], chaderins [123,124], or claudins [125,126]; synaptic proteins [127,128,129]; and channels and pores, for which connexins and NMDAR receptors are the most relevant [130]. Eph receptors and ephrins also interact with proteases on the cell surface, such as ADAM10, that can cut ephrins at the membrane level, thus ending ephrin signaling [131]. Importantly, many of these signaling pathways are activated simultaneously to achieve a given output, and the outcome can be either agonistic or antagonistic, depending on the cellular context.

### 3.1. Neuronal Eph–Ephrin System and Pain

In the last 30 years, increasing evidence has confirmed the involvement of the Eph–ephrin signal pathway in the modulation of pain. Several ephrin ligands and receptors are expressed by sensory neurons in the superficial laminae of the spinal cord and the DRGs, mostly in the small and medium neurons, and by glial cells.

They localize mainly at synapses, at both pre- and postsynaptic levels, where they regulate numerous developmental and functional processes. Since EphB–ephrinB signaling regulates spinal sensory connectivity, it was suggested to modulate pain [132]. Interestingly, it has been hypothesized that they mediate or contribute to activity-dependent alterations at the synapses.

Synaptic plasticity is well recognized as a mechanism underlying chronic pain. Synapses along the nociceptive pathway can alter their strength in an activity-dependent manner; these changes are both structural and functional and occur at both pre-synaptic and post-synaptic levels [133]. Involvement of the NMDAR, specifically when the NR2B subunit is altered, is considered a critical mechanism underlying synaptic plasticity at the SDH in the context of chronic pain, as it orchestrates the development of LTP. Interestingly, EphB2–ephrinB2 signaling has been shown to modulate synaptic plasticity in the hippocampus via interacting with the NMDAR [134]. Furthermore, Battaglia and colleagues demonstrated that EphB1–ephrinB1 signaling modulates synaptic efficacy in an NMDAR-dependent manner in the spinal cord [132]. In particular, EphBs phosphorylate NMDARs via the src family [135], and the application of an src kinase inhibitor blocks EphB1-dependent phosphorylation of the NR2B subunit and thermal hyperalgesia [135]. Moreover, a binding assay shows that the interaction site between the EphBs and NMDARs is extracellular [130]. The extracellular domain of EphBs 1–3 interacts directly with NMDARs via the specific tyrosine residue Y504, which is important for targeting and retention of NMDARs at synapses [130,136]. Injury-dependent phosphorylation of Y504 seems necessary and sufficient to bind NMDAR, increasing the receptor’s affinity [136]. EphB-dependent phosphorylation of NMDAR leads to an increased calcium influx through NMDARs, activation of Ca^2+^-dependent kinases, and alteration of gene expression [134]. This first evidence suggests the key role of the Eph–ephrin system in the physiology of the spinal cord and its contribution to pain modulation.

EphBs are involved in the development and plasticity of excitatory synapses, also through interaction with AMPA glutamatergic receptors and NMDARs [127,130,134,137]. Song et al. showed that EphB–ephrinB signaling is required for LTP of synapses between DRG neurons and dorsal horn neurons, pointing out the importance of this signal pathway in the synaptic plasticity of pain pathways [138]. Indeed, blocking the EphB receptor suppresses the hyperexcitability and abnormal spontaneous activity of both DRG neurons, the wide dynamic range of SDH neurons produced by neuronal damage [139,140,141]), and thermal hyperalgesia and mechanical allodynia [138,142].

To date, the importance of EphB–ephrinB signaling as a mechanism to mediate physiological pain and chronic pain, including neuropathic pain, is quite well demonstrated (Figure 2). Alterations in the receptors and ligands expression at the level of the spinal cord and/or DRGs have been described in several mouse models of pain and in patients, and stimulations of Eph receptors expressed by spinal neurons via the ephrinB2-Fc fragment is sufficient to induce thermal hyperalgesia in an src kinases-dependent manner by phosphorylation of NMDAR [132,135]. Mice with deleted ephrin-B2 in Na_v_1.8 positive nociceptive sensory neurons in DRGs show reduced pain behavior in the complete Freund’s adjuvant (CFA)-induced inflammatory pain model, formalin-induced pain, and in a model of neuropathic pain, without affecting acute pain behavior and motor coordination [143]. Furthermore, these transgenic mice show diminished tyrosine phosphorylation of NMDA receptors in the dorsal horn, along with reduced c-fos expression after CFA injection, suggesting that ephrinB2 signaling plays a crucial role in regulating pain thresholds after pain induction [143]. In a model of neuropathic pain, the level of ephrinB2 was upregulated in the DRG and spinal cord in a time-dependent way, and its knockdown was sufficient to reduce injury-induced mechanical allodynia [144], pointing out a new role for the EphB2–ephrinB2 system as a modulator of the neuronal network underlying chronic pain. Indeed, in DRG, the ephrinB1 gene is upregulated after activation of the lysophosphatidic acid receptor 1 (LPA1) receptor and downstream Ras homolog gene family member A (RhoA) [145]. Along the same lines, downregulation of ephrinB1 by antisense oligodeoxynucleotide abolishes LPA-induced pain behavior, whereas activation of EphB with ephrinB1-Fc induces pain behavior resembling neuropathic pain [146]. It has been suggested that the contribution of EphB–ephrinB signaling to the development of neuropathic pain following neuronal damage is mediated by synaptic plasticity modulation between sensory neurons of the DRGs and dorsal horn nociceptors [142].

EphB–ephrinB signaling is also involved in various models of inflammatory pain. Persistent inflammatory pain can be efficiently prevented and treated by blocking spinal EphB–ephrinB signaling [132]. Inflammatory pain is partially mediated by induction of COX-2 expression in the spinal cord [147]. Interestingly, intrathecal injection of ephrinB2-Fc increases Cox-2 levels and pain behavior, whereas inhibition of Cox-2 prevents pain behavior induced by ephrinB2-Fc. In agreement, EphB-Fc injection reduces CFA inflammatory pain and decreases Cox-2 expression [148]. EphB2 and ephrinB2 are upregulated in the enteric nervous system, especially in the colonic nerves in patients with irritable bowel syndrome (IBS) [149], and in a rat model of IBS where the intensity of visceral hypersensitivity characteristic of IBS correlated positively with the upregulation of ephrin signaling [149]. The immediate early genes *c-fos* and *arc* are considered markers of synaptic rearrangement [150], as they correlate with neuronal activity [151] and with cytoskeleton rearrangement at the postsynaptic level [152], respectively. Their upregulation in the colons of IBS patients and rats suggests increased synaptic plasticity at the colonic enteric-nervous-system level. In support of this hypothesis, increased synaptic densities and expression of associated proteins such as PSD-95 are observed. As a mechanism of action, it has been proposed that EphB2 induces src-dependent phosphorylation of NR2B, increasing Ca^2+^ permeability of the channel, thereby inducing upregulation of c-fos and arc. This hypothesis was supported by the observation that blockading of the NMDARs-dependent Ca^2+^ influx reduces IBS-dependent hypersensitivity [149]. In another model of IBS, EphB2–ephrinB2 signaling was held responsible for myenteric synaptic plasticity and subsequent visceral hypersensitivity, since it mediates neurite outgrowth and sprouting [153,154]. Interestingly, downregulation of the EphB6 receptor has been reported in a model of colitis [155]. The kinase domain of this receptor is non-functional, so its function is to sequester the ligands and reduce forward signaling. Therefore, its decrease results in an increase in the forward signal’s strength.

EphBs–ephrinBs signaling has also been associated with cancer pain. In a model of bone cancer pain, Eph–ephrin was associated with the maintenance of mechanical hypersensitivity through modulation of the expression of pro-inflammatory cytokines such as IL-6, IL-1β, and TNF-α at the level of the spinal cord [156]. The analgesic effect of the compound Z-360 was evaluated using another model of pancreatic-cancer-induced pain [157]. At the level of DRGs, this molecule blocks the release of IL-1β from the inoculated tumor, preventing ephrinB1 upregulation and failing to phosphorylate NR2B. Liu et al., using a bone cancer pain model, demonstrated the importance of the EphB1 receptor in cancer-dependent hypersensitivity and the development of morphine tolerance. Indeed, blocking or genetic deletion of EphB1 prevents and reverses cancer pain and morphine tolerance [158,159]. Indeed, the IL-1β/ephrinB1/NR2B axis has been proposed to underlie the development of opioid resistance [160].

Other forms of pain have been associated with activation of the EphB–ephrinB signaling pathway. In the context of diabetes, EphB1 seems to be more involved in the maintenance of pain than in its development [161]. In the STZ model, upregulation of the phosphorylated form of EphB1 is associated with the activation of astrocytes and microglia [161]. Moreover, the repetitive blockade of EphB1 receptor by infusion of EphB1-Fc reduces DNP, gliosis, and pro-inflammatory cytokine release. In addition, EphBs have been associated with opioid-induced analgesia. The drug Remifentanil, a synthetic opioid analgesic drug, induces ephrinB, EphB1, and c-fos in dorsal horn neurons, and leads to the development of opioid-induced hypersensitivity, which is prevented by blocking EphB–ephrinB signaling [162].

Since the EphB–ephrinB system is recruited in different models of chronic pain, ranging from inflammation to cancer-associated pain and neuropathic pain, the activation of the Eph–ephrin signal seems to be a crucial mechanism that is common to multiple forms of pathologic pain. Cibert-Goton et al. showed how this system is activated by stimuli of different origins (inflammation or neuronal damage), but leads to the same result, i.e., the involvement of the EphB–ephrinB system. In EphB1-KO mice, while acute pain is unchanged, chronic pain of different origins is impaired. Specifically, the lack of activation of the EphBs system leads to decreased phosphorylation of NR2B, resulting in reduced Ca^2+^ entry and neuronal activity [163]. Surprisingly, EphB1-KO mice show also decreased microglial activation, most likely due to the reduction of neuronal activity [164].

In the context of pain transmission, the forward signal has been better characterized. EphBs activation induces the activation of several downstream signal pathways involved in increasing nociceptor excitability at the level of the dorsal horn, and synaptic plasticity—crucial mechanisms underlying chronic pain (reviewed in [165]). MAPKs, phosphoinositide 3-kinase (PI3K), PKCγ, and PKA are all involved in different chronic pain models in an NMDAR-dependent manner [166,167,168,169,170,171]. Activation of the EphB receptor by intraplantar or intrathecal injection of ephrinB1-Fc induces hyperalgesia and activation of MAPKs, including p38, JNK, ERK, the PI3K-AKT pathway, PKCγ, and the PKA pathway, both peripherally and/or centrally, depending on NMDAR activity [166,167,168,170,171,172,173,174]. Furthermore, blocking Eph–ephrin signaling in different contexts of chronic, inflammatory, neuropathic, and cancer-associated pain by injection of EphB1-Fc attenuates, not only thermal hypersensitivity and mechanical allodynia, but also the activation of p38, JNK, and ERK [167,175], the PI3K-Akt pathway [168], and PKCγ and PKA [172,174]. Interestingly, inhibition of PI3K upon stimulation of EphB counteracts EphB-dependent pain behavior and activation of ERK [168,173], indicating crosstalk between the two-kinase systems [169]. Importantly, PKCγ-KO mice develop less pain following spinal activation of the EphB receptor [172]. The exact mechanism of PKA activation is still unclear, but it was suggested that upon EphB activation, PKA may be activated through a mechanism involving the NMDA-dependent increase in Ca^2+^ transients increasing cAMP [172,174], or through a more complex mechanism involving the release of pro-inflammatory cytokines by glial cells [176].

Even if the involvement of all these signaling pathways is well established, the specific contribution of each effector to the outcome in the different pain conditions and whether there is a diverse signaling contribution of each pathway in different forms of chronic pain are not well understood and need further study.

### 3.2. Eph-Ephrin System and Glia in Pain

The contribution of the Eph–ephrin system to the development and maintenance of chronic pain has been well studied at the neuronal level; however, little is known about the involvement of the glia’s ephrin system in pain transmission (Figure 2).

Astrocytes express a wide variety of ephrins and Eph receptors [177,178], which are regulated following neuronal damage [179,180]. EphrinA5 is considered a marker of astrogliosis, and EphA3 is selectively upregulated in reactive astrocytes after brain injury [181]. While in neurons the communication between EphBs and ephrinBs is most often studied, the focus of astrocyte–neuron crosstalk is on the interaction of class A receptor-ligands. Indeed, the EphA–ephrinA signaling strongly regulates the neuron–astrocyte interaction modulating functional and structural plasticity [178]. The interaction between the EphA4 receptor localized on dendritic spines and ephrinA3 on astrocytes modulates the development of certain forms of LTP in the hippocampus through the regulation of glutamate transporters on astrocytes, consequently regulating the concentration of glutamate [182,183]. This interaction between dendritic EphA4 and astrocytic ephrinA3 is also involved in chronic inflammatory pain in mice [184], emphasizing how LTP and pain-related central sensitization share basic mechanisms [185]. EphA4 is upregulated following inflammatory stimuli [184] and neuronal damage [186], and blocking EphA4 impairs the development of chronic inflammatory pain [184] or neuropathic pain [186]. Interestingly, in a model of trigeminal neuropathy, EphA4 increases in reactive astrocytes, and its blockading leads to pain relief, whereas in a model of spinal cord injury (SCI), EphA4 increases in both neurons and astrocytes, and its blockade increases pain, indicating the complementary role of forward and reverse signaling in different cell types. Furthermore, it was suggested that in SCI conditions, EphA4 has a protective role in blocking the sprouting of sensory fibers, working as an inhibitor of axon growth [187].

The glial EphB–ephrinB system has also been implicated in pain transmission. Several studies report activation of microglia and astrocytes as a result of the EphB–ephrinB signaling activation in chronic pain conditions. For example, Liu et al., in a cancer-associated pain model (TCI), showed increases in EphB1 and toll-like receptor 4 (TLR4) on glia, along with gliosis and enhanced release of pro-inflammatory cytokines [176]. Individual blockading of each receptor reduced gliosis, the concentrations of IL-1β and TNF-α, and cancer-associated hypersensitivity; the activation of the EphB1 receptor itself induced both gliosis and thermal hypersensitivity. In another study, Erk5 and cAMP response-element-binding protein (CREB) activation were mostly found in microglia following neuronal damage [188]. Furthermore, in the STZ-induced diabetes model, while increased phosphorylation of EphB1 expressed by glia positively correlates with gliosis and neuropathic pain [161], the blockade of EphB1 leads to decreased astrocytosis and cytokine release.

This evidence points out the Eph–ephrin system as a new potential target for developing new pain therapies.

## 4. Semaphorin–Plexin System

Semaphorins (Semas) constitute a large family of highly conserved signaling proteins expressed in the majority of tissues. Five classes of Semas have been identified in vertebrates. Semas belonging to class 3 are secreted; the members of class 7 are anchored to the membrane by a GPI-tail; and the members of classes 4, 5, and 6 are transmembrane proteins and are released extracellularly [189]. Though discovered as important axon guidance molecules during development, now we know that Semas play an essential role in several different physiologic systems, participating in a wide amount of processes spanning from embryogenesis to adult tissue homeostasis [190]. Indeed, they contribute to cardiomyogenesis [191,192]; osteoclastogenesis [193]; angiogenesis [194,195,196]; functioning of nervous, endocrine, respiratory, and musculoskeletal systems; and immunomodulation [197,198,199], among others. Furthermore, they are involved in diseases affecting these systems and in cancer progression, specifically in tumor neovascularization and metastasis [200,201,202].

Semas can bind several different protein families that function as receptors and transmit their signals. The most known Sema receptors are plexins (Plxns). Plxns are large single-pass transmembrane proteins subdivided into four classes (A–D). They show a variety of activation mechanisms, such as ligand-dependent dimerization and conformational changes. They are characterized by an extracellular sema domain that binds Semas by a highly conserved intracellular domain containing a GTPase activating protein (GAP) homology domain. Interestingly, each Plxn shows a preferential affinity with a given Sema subclass [203]. In different cell types, activation of the Sema/Plxn signaling leads to morphologic changes (affecting actin and microtubule cytoskeletons) and reduced cell adhesion. The most important mediators of Sema/Plxn signaling are the small GTPases, well-known regulators of the cytoskeleton and cellular adhesion promoting integrin functions [204]. All Plxns can directly activate the GTPase activity of the Ras and Rap family thanks to a highly conserved intracellular GAP homology domain [205]. Activation of Plxns and its GAP activity leads to reduced integrin activation towards lower levels of active R-Ras (GTP-bound form). Plxns, through a Rho binding domain (RDB), interact also with Rho family GTPases [206], crucial elements for the control of cell shape and movement.

Neuropilins (Nrp1/2) are transmembrane proteins that serve as co-receptors for the secreted class 3 Semas (Sema3s). Nrp1/2 has a short cytoplasmatic domain that is not required for signal transduction. Frequently, Nrps only stabilize the interactions between Semas and the receptors. Indeed, to transduce the intracellular signal, Nrps are associated with other proteins, such as Plxns. Interestingly, other membrane proteins, such as CD72 [207], Tim2 [208], integrins [209], and proteoglycans [210], can directly bind Semas.

Further complexity in Semas signaling is given by their co-receptors and associated proteins. A large variety of molecules associated with Sema–Plxn complexes, working as co-receptors, profoundly influence the signaling outcomes. In addition, several RTKs and cytoplasmatic tyrosine kinases, such as vascular endothelial growth factor receptor 2 (VEGFR2), Met, ErbB2, Src, and Fyn, among others, associate with Plxns or Nrps, and can dramatically alter the outcomes of signaling, becoming transactivated while being phosphorylated in a ligand-independent manner upon Sema binding [211]. Specific plexins can associate with different tyrosine kinase receptors, eliciting divergent functional outcomes. Moreover, transmembrane Semas can also act as receptors [212], starting reverse signaling in Semas-expressing cells, as seen for Ephrins.

Therefore, depending on the cellular context, semaphorins might trigger multiple signaling pathways, mediating different and occasionally opposing functional effects.

### Semaphorin–Plexin Signaling in Pain

Sema–Plnx signaling was originally discovered as repulsive axon guidance molecules [213]. However, over the past few years, Semas have been shown to be involved in many other developmental processes that shape the CNS and PNS (reviewed in [214]).

Less is known about the role of this system in the physiology of the adult nervous system. Many Semas have been seen to play a crucial role in different aspects and functions of the adult CNS. In particular, they are implicated in the inhibition of neurogenesis (Sema3A and Sema7A) [215,216], re-innervation of taste receptors (Sema3A, Sema7A) [217], maintenance of hippocampal synaptic connectivity, retention of fear memories (Sema3G, Sema4C) [218,219], and the functioning of corticostriatal circuits (Sema3F) [220]. In addition, Semas play a crucial role in maintaining homeostatic synaptic plasticity and controlling hippocampal synaptic transmission (Sema3F) [221,222]. Therefore, these new findings point out a role for semaphorin signaling in the regulation of neuroplasticity.

As synaptic plasticity is a well-recognized mechanism underlying chronic pain [133], it is tempting to speculate that the Sema–Plxn signaling pathway may contribute to the development and maintenance of chronic pain. Compared to the other signaling systems involved in neuronal development, little is known about the role of Semas in chronic pain, but evidence has been recently reported on the involvement of these guidance molecules in pain (Figure 3).

During the development of the nervous system, Sema3A prevents axons from innervating inappropriate territories [223,224]. In particular, it has been shown that Sema3A repels axons from a subset of small diameter, nerve growth factor (NGF)-responsive embryonic DRG neurons that are involved in thermoreception and nociception [225,226,227]. The expression and secretion of Sema3A and the expression of Nrp persist in the adult nervous system [228] and are upregulated at the injury site of the sectioned spinal cord, where it inhibits regeneration of nerve fibers and restoration of neural circuitry [229,230]. Inhibition of Sema3A induces re-connection of transected axons of the spinal cord and restores motor function [231]. Moreover, overexpression of Sema3A prevents the sprouting of unmyelinated sensory nerve endings and attenuates hyperalgesia in the spinal cord of the NGF-induced neuropathic pain model [232] and in the injured cornea [233]. Interestingly, in the CCI model of neuropathic pain, intrathecal injection of Sema3A reduces mechanical allodynia and thermal hyperalgesia, and partially restores the decrease in IB4-positive non-peptidergic unmyelinated sensory nerve terminals in lamina II of the dorsal horn. Furthermore, Sema3A does not alter the sprouting of myelinated nerve terminals [234], suggesting an anti-nociceptive effect of Sema3A.

Altered semaphorin levels have been detected in several chronic inflammatory diseases associated with reduced noradrenergic innervation, such as endometriosis or rheumatoid arthritis (RA); psoriasis; Crohn’s disease; and immunometabolic diseases such as obesity, diabetes, and atherosclerosis, characterized by chronic tissue inflammation [235]. In chronic inflammatory diseases, noradrenergic hypo-innervation correlates with the progress and severity of the disease [236], whereas peptidergic innervation is significantly increased in peritoneal endometriosis and RA. These alterations lead to an imbalance in anti- or pro-inflammatory neurotransmitters thought to maintain a chronic inflammatory milieu [237].

Reduced noradrenergic innervation associated with increased expression of Sema3C and Sema3F has been found in tissues from patients with pelvic endometriosis, whose main symptom is pain. Significantly increased content of macrophages is found in peritoneal fluid and tissue of endometriosis patients [238,239]. Interestingly, Semas are expressed by lesion-associated macrophages and fibroblasts while Nrp and PlexinA receptors are present in nerve fibers [237]. Different studies have revealed a role of semaphorins in the innervation changes observed during the progress of diseases such as RA and Morbus Crohn [240], suggesting that neuroimmunomodulatory processes might be responsible for such changes in endometriosis since it is known that innervation can be modulated by immune cells [241].

Pathological innervation associated with altered expression of class 3 Semas including Sema3A, Sema3C, and Sema3D has been suggested as a mechanism underlying chronic low back pain [242]. In particular, Sema3A has been proposed as a candidate target against low back pain as a potential mechanism for its pathogenesis [243,244]. Low back pain is often associated with degeneration of intervertebral discs. Under physiological conditions, innervation does not penetrate the discs leaving the intravertebral discs avascular and aneural. On the contrary, in degenerated discs has been observed a strong growth of nociceptive nerve fibers and blood vessels, which may contribute to pain [245,246]. In the healthy disc, Sema3A is highly expressed and localized in the outer annulus fibrosus, whereas in degenerated specimens Sema3A expression is significantly decreased in this region, and it appears as a good candidate for low back pain treatment.

Recently, the signaling Sema3B/PlxnA1 and Sema3B/PlxnA2/Nrp2 have been involved in the pathophysiology of RA, both in patients and in a mouse model [247,248]. Sema3B amount was reduced in the synovium of patients with early RA and its expression level correlates inversely with the expression of inflammatory mediators and clinical manifestations [247]. Furthermore, genetic knockdown of Sema3B induces higher arthritis severity together with higher expression of cytokines, chemokines, and matrix metalloproteinase. This effect is mainly due to fibroblast-like synoviocytes that also have an increased migratory capacity, consistent with the invasive and aggressive phenotype of RA. Interestingly, arthritic mice show also a reduced expression of PlexinA2 and the co-receptor Nrp-1, a receptor complex that binds Sema3B, Sema3A, and Sema3F which are known to have a protective role in the pathogenesis of RA [248]. The protective role of Sema3B is likely mediated by inhibition of the ERK pathway. In fact, ERK is overactivated in the joints of Sema3B^-/-^ arthritic mice [248] and synovial tissue from patients with RA and from patients with early arthritis who develop erosive RA [247]. In contrast, a reduction in ERK activation has been found in Sema3B-stimulated RA fibroblast-like synoviocytes [247]. All these reports support a protective role of class 3 semaphorin in the pathogenesis of RA. Since the clinical parameters of patients correlate positively with Sema3B expression we speculate that also RA-related pain behavior will be improved by Sema3B; however, a direct experimental proof of a direct link is still missing.

Interestingly, while in neuropathic pain models Sema3A has an analgesic role, in a bone tumor-associated pain model it has a pro-nociceptive effect [249]. Indeed, using a model of bone metastasis where tumor cells are inoculated into the femoral bone, they demonstrated an increase in Sema3A presumably produced by the inoculated cells. Knocking down Sema3A results in a decelerating of cancer cell proliferation and improved pain behavior [249]. Normally, bone metastases induce the sprouting of sensory nerves innervating the bone [250,251] associated with pain [252]. Since Sema3A has a repulsive effect on sensory fibers, one would expect that blocking this signal would have a beneficial effect on pain. However, considering tumor cell proliferation with increased intraosseous pressure and bone resorption resulting in cytokine release [253,254], predisposes the final effect of Sema4C toward a pro-nociceptive action [249].

Class 3 Semas and the Sema3s/PlexinA/Nrp signaling have been frequently associated with anti-nociceptive function. However, more recently our laboratory demonstrated the involvement of Sema4C-PlexinB2 signaling in modulating inflammatory nociceptive hypersensitivity. Indeed, we showed, using an inflammatory pain model, that a developmental important system is rekindled in adult life to mediate nociceptive hypersensitivity by promoting both acute and long-lasting plasticity of sensory neurons [255]. We found that CFA injection into the paw induces upregulation of PlexinB2 and Sema4C into the DRG neurons and of Sema4C by keratinocytes and immune cells infiltrating the inflamed skin (macrophages and T-cell) pointing out the importance of the crosstalk between neurons and immune cells in inflammatory pain transmission. PlexinB can mediate very complex intracellular signaling, activating different molecular pathways depending on cell type, developmental stage, or cellular context. Accordingly, in early embryonic developmental PlexinB2 functions are entirely mediated by the Ras GAP domain [256], whereas in adult neurons the receptor recruits RhoA-ROCK signaling to promote the increase in TRPA1 ion channels in the cell membrane, thus sensitizing DRG sensory neurons [255]. It is tempting to speculate that this system could participate also in other forms of chronic pain such as neuropathic or cancer pain. Indeed, several kinds of cancer cells express and can release Semas, and some neuropathic pain conditions have a strong inflammatory component.

PlexinC1-Sema7a signaling mediates an acute inflammatory response [257]. PLXNC1 genetic depletion (*PLXNC1^−/−^* mice) or anti-PlexinC1 antibody treatment results in a reduced inflammatory response and lower cytokine and chemokine production in vivo, letting us hypothesize a modulation of inflammation-associated pain as well, highlighting the importance of the awakening of a developmental crucial system in pain transmission.

More recently it was shown the involvement of PlexinD1 in neuropathic pain transmission. Specific autoantibodies against antigens in the somatosensory pathway are recognized as novel mediators of neuropathic pain [258,259]. Anti-PlexinD1 antibodies were discovered during a serum screening which aimed to identify autoantibodies that specifically bound sensory neurons in the DRG and SDH [260]. Anti-PlexinD1 antibodies are found in a small portion of patients with neuropathic pain and underlying neuroinflammatory diseases [260], painful trigeminal neuropathy [261], and small fiber neuropathy [262]. Interestingly, immunotherapies ameliorate neuropathic pain in patients´ anti-plexinD1 positive and passive transfer of IgG purified from these patients to mice induces mechanical and thermal hypersensitivity [261,262].

Anti-PlexinD1 antibodies bind mainly with IB4- and P2X3-positive neurons, in the DRG and lamina I and II of the dorsal horn, and VIP-positive parasympathetic nerve fibers in the skin. In vitro studies show that the binding of anti-PlexinD1 antibodies increases the membrane permeability of DRG neurons and induces cellular swelling without complement activation [260]. As mechanism, it is alleged that anti-PlexinD1 antibodies may induce DRG neurons´ cytotoxicity through the dysregulation of cytoskeleton stability. Nevertheless, the causal link between anti-PlexinD1 antibodies and neuropathic pain needs further experimental proof.

Semaphorins as guide molecules during development are known to play a dual role in both axon repulsion and attraction. Interestingly, a duality is also maintained in their function in the adult organism, particularly in the context of chronic pain. Indeed, depending on the cellular context in which they are activated, they can exert both a pro-nociceptive function and mediate analgesia.

## 5. ncRNAs and Axon Guidance Molecules in Pain

Recent literature has investigated the role of non-coding RNAs (ncRNAs) in pain. Many studies identified changes in the expression of various ncRNAs in patients affected by chronic pain and in chronic pain models, demonstrating that the dysregulation of different ncRNAs promotes or inhibits the occurrence and development of chronic pain [263]. Among ncRNAs, microRNAs (miRNAs), small non-coding RNAs 21–23 nucleotides in length that play key roles in modulating gene expression at the post-transcriptional level [264], are widely reported to be involved in neuropathic pain. Interestingly ncRNAs are shown to regulate also Wnt, semaphorin, or ephrin pathways in the contest of pain. Using the CCI model of neuropathic pain, it is demonstrated the importance of miRNAs in regulating Wnt pathway and pain. Indeed, the downregulation of different miRNAs is correlated to activation of the Wnt pathway and increased mechanical and thermal hyperalgesia [265,266,267]. Overexpression of miR24-3p, miR216-5p, miR146a-5p, or miR30b-5p attenuates inflammatory cytokines release, mechanical allodynia, and thermal hyperalgesia [265,266], and reduces the level of the Wnt pathway-related gene (β-catenin, c-myc, and cyclin D1 [266] inactivating the Wnt/β-catenin signaling pathway [265], or negatively regulating Wnt5a [268].

Interestingly, the miRNA miR-30b-5p is also reported to target SEMA3A in the model of traumatic brain injury (TBI) [269] and spinal cord injury (SCI) [270]. The agomir of miR-30b (a double-stranded RNA that mimics the endogenous miRNA) can regulate Sema3A/NnpP-1/PlexinA1/RhoA/ROCK axis in vivo and restore spinal cord sensory conductive function [270]. It is tempting to speculate a role of the miR-30b-5p as a regulator of SEMA3A also in chronic pain models.

MiRNAs can be regulated by other ncRNAs, such as the long non-coding RNAs (lncRNAs) that can bind miRNAs suppressing their action. The lncRNA colorectal neoplasia differentially expressed gene (CRNDE) has been shown to mediate neuropathic pain progression in the CCI model of pain serving as a sponge for miR-146a-5p thus increasing Wnt5a pathway [268]. Whereas the silencing of CRNDE attenuates mechanical allodynia, thermal hypersensibility, and the inflammatory response in vivo, showing a lncRNA CRNDE/miR-146a-5p/Wnt5a axis [268]. It has been shown that the Eph–ephrin signaling pathway is regulated by lncRNAs. In CCI rats, the ultraconserved lncRNA uc.153 level is increased in the spinal cord and its knockdown prevents CCI-induced pain behaviors. Uc.153 negatively modulates Dicer-mediated pre-miR-182-5p processing and inhibits its maturation [271]. Moreover, spinal miR-182-5p downregulation increases the expression of EphB1 and p-NR2B (phosphorylated NR2B), facilitating hyperalgesia [271,272].

Interconnection between Wnt and Semas pathways has been demonstrated also at the level of ncRNAs. Intervertebral disc degeneration (IDD) is considered a significant contributor to low back pain. In IDD specimens the expression level of circular RNA SEMA4B (circSEMA4B) is reduced in nucleus pulposus cells (NPCs). One of the targets of circSEM4B is miR-431, which negatively regulates the secreted frizzled-related protein 1 (SFRP1) and GSK-3β, two inhibitors of Wnt signaling [273]. CircSEMA4B acts as a sponge for miR-431 and thus regulates the level of SFRP1 or GSK-3β, which in turn inhibit Wnt signaling, reducing the IL-1β-induced degenerative process in NPCs. Therefore, rescuing circSEMA4B expression in NPCs may be a prospective approach for improving IDD [273].

Despite recent progress in studying the regulation of Wnt, Eph–ephrin, and Sema-Plxn signaling pathways by ncRNAs [274,275,276] and their contribution to pain modulation, this area deserves further investigation.

## 6. Conclusions

Despite the attempts by many laboratories, the precise spatiotemporal sequence of activation of the different pathways in the context of pain is not entirely clear. The knowledge of which pathway is activated at a precise moment in a particular cell type will help to develop a target-oriented pharmaceutical approach against pain more specific and thus with fewer side effects. As we have seen, each class of axon guidance molecules described in this review is capable of activating a plethora of intracellular cascades often overlapping or leading to opposite effects. For example, Sema3A mediates nociception in the context of cancer-associated pain [249] while it has an analgesic effect in the context of neuropathic pain [234]. Similarly, identifying the downstream effectors supports the possibility of reducing the side effects of new drugs and making them more effective. Indeed, one must consider that these pathways interact with others that are activated at the same time by other mediators and are modulated by them. Thus, targeting a pathway too far upstream may give dangerous effects or none at all. Indeed, axon guidance molecule signaling often regulates tissue homeostasis. Moreover, different mediators can activate the same effector using different pathways. Wnt3a and Sema4C mediate mechanical hyperalgesia in contexts of cancer-associated or inflammatory pain respectively. In both cases, the downstream effector is an increased membrane availability of TRPA1 expressed by sensory neurons in DRGs. Interestingly, this is achieved by different pathways: Wnt activates the Rac-JNK pathway, whereas Sema4C engages the RhoA-ROCK pathway.

The interaction of guidance molecules with other pain mediators must also be considered from a therapeutic perspective. For example, several Wnt ligands have been shown to induce the release of pro-inflammatory cytokines in the spinal cord [28]. In particular, Wnt5a in the context of HIV-associated pain induces the release of BDNF [277], a factor that induces the disinhibition of spinal GABAergic neurons [48,278].

Pro-inflammatory processes modulate the expression of axon guidance molecules and these changes regulate the onset and resolution of neuroinflammation. Different families of axon guidance molecules have been shown to regulate neuroinflammation modulating glia functions, but the precise mechanisms underlying cell-to-cell interactions or intracellular signaling are not well understood [279]. Understanding the ligand-receptor combinations present and their specific roles will be the key to figure out the regulatory functions of axon guidance molecules in the context of neuroinflammation and may help to discover new molecular targets to treat chronic pain.

Interestingly, both Wnt and Eph signaling have been implicated in side effects due to opioid administration. In particular, inhibition of Wnt5a ameliorates the exacerbation of HIV-related neuropathic pain, induced by continuous morphine administration [46]. Furthermore, the blockade of EphB1 upregulation prevents the development of tolerance to opioids in a cancer pain model [158]. This suggests that the combined therapies targeting more than one molecule will be more successful than drugs focusing only on one signaling pathway.

Despite many approaches and molecules developed to target axon guidance molecules-activated pathways appear to be efficient in pre-clinical models of pain, very few of them succeed in phase 1 or 2 of the clinical tests. This underlies that a lot of work has still to be performed to deeply understand this complex and paradoxical signaling system. To develop more efficient and safe drugs or therapies, many questions lack a complete answer: (i) Do these molecules and their interaction play the same role in mice and humans? (ii) Although the pre-clinical studies mainly focus on the spinal effect, what is the role of these molecules at the brain level? (iii) Seen the strong involvement of microglia, what role do these axonal guidance molecules take regarding the sex and gender differences?

In conclusion, attention should be directed towards reinforcing the resolution process by increasing the expression of anti-inflammatory endogenous regulators such as anti-inflammatory cytokines, resolvins, protectins, and maresins, and modulating the interaction at the level of ncRNAs.

## Figures and Tables

**Figure 1 cells-11-03143-f001:**
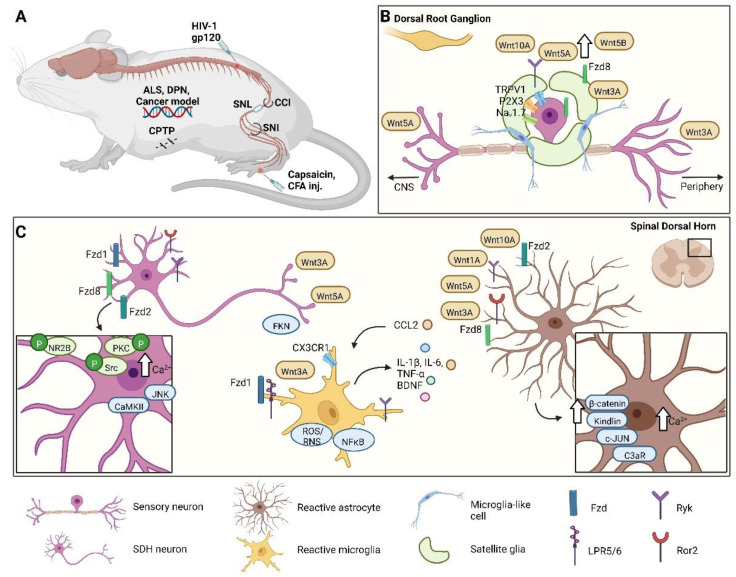
**Wnt’s contributions to different pain conditions in various neuronal and non-neuronal cells.** (**A**) Schematic representation of different pain models in mice. (**B**) After neuropathic pain induction, in dorsal root ganglia, activated nociceptors increase the receptors’ expression, such as TRPV1, P2X3, Na1.7, Ryk, and Fzd8, and that of Wnt-related proteins, e.g., Wnt5a, Wnt5b, Wnt3A, and Wnt10a. Likewise, in satellite cells, Fdz8 and Wnt3a are also overexpressed. (**C**) In the spinal dorsal horn, Wnt signaling is involved in pain sensation and acts on neuronal and non-neuronal cells. Many Fzd receptors and co-receptors (Ryk and Ror2) are upregulated in neurons, which brings about phosphorylation, activation of downstream targets (NR2B, Src, and PKC), and a Ca^2+^ increase. This results in increases in JNK and CaMKII and the release of Wnt3a, Wnt5a, and FKN. Astrocytes harbor large amounts of Wnt proteins and receptors, which are upregulated during pain. As a consequence, the concentration of Ca^2+^ rises in the cytoplasm, along with the concentrations of β-catenin, kindlin, c-JUN, and C3aR; and the release of CCL2 increases. This last chemokine triggers microglia, which in situations of pain upregulates Fzd, the LRP5/6 co-receptor, and Wnt3a. Reactive microglia increase the expression of CX3CR1, ROS/RNS, and NF-kB and the secretion of IL-1β, IL-6, TNF-α, and BDNF, which in turn escalate the inflammatory condition. ALS, amyotrophic lateral sclerosis; DPN, diabetic peripheral neuropathy; CPTP, chronic post-thoracotomy pain; CCI, chronic constriction injury; SNL, spinal nerve ligation; SNI, spared nerve injury; CFA, complete Freund’s adjuvant; inj., injection; Fzd, Frizzled; FKN, Fractalkine; P, phosphorylation; NR2B, N-methyl-D-aspartate receptor subunit 2B; JNK, c-Jun amino (N)-terminal kinase; Src, Proto-oncogene tyrosine-protein kinase; PKC, protein kinase C; CaMKII, Ca^2+^/calmodulin-dependent protein kinase II; CCL2/MCP1, CC-chemokine ligand 2; ROS/RNS, reactive oxygen species/reactive nitrogen species; NF-kB, Nuclear factor kappa-light-chain-enhancer of activated B cells; BDNF, brain-derived neurotrophic factor; c-JUN, transcription factor Jun; C3aR, complement component 3 fragment a receptor; LRP5/6, low-density lipoprotein receptor-related protein 5/6; Ryk, receptor-like tyrosine kinase; Ror2, receptor tyrosine kinase-like orphan receptor 2; CX3CR1, CX3C chemokine receptor 1; IL, interleukin; TNF-α, tumor necrosis factor alpha.

**Figure 2 cells-11-03143-f002:**
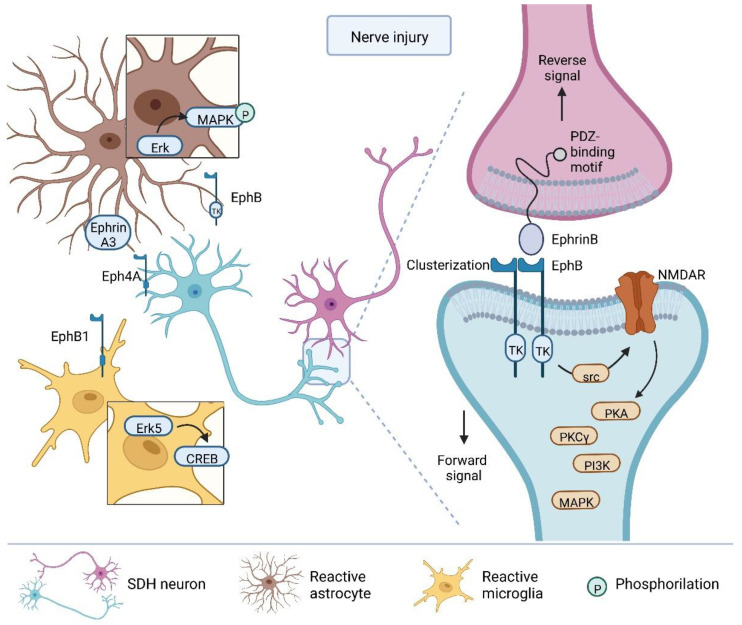
**Eph/ephrin signaling in nerve injury causes overactivation of nociceptors, glial cells, and synaptic plasticity.** EphB receptors belong to a family of RTKs and bind the membrane-bound ligand ephrinB. In pain states, this interaction helps the polymerization of the receptor and amplification of the forward signal towards overexcitability of sensory neurons (via NMDAR) and modification of synapses. In chronic inflammatory pain, the dendritic EphA4 is upregulated following neuronal damage, interacts with ephrinA3 (astrocytic), and activates its cascade involving ERK and activation (phosphorylation) of MAPK. After injury, Erk5 and CREB are activated, possibly due to EphB1 triggering. TK, tyrosin kinase; Erk, extracellular signal-regulated kinase; MAPK, mitogen-activated protein kinase; CREB, cAMP response element-binding protein; NMDAR, N-methyl-D-aspartate receptor; PKA, protein kinase A; PKC γ, protein kinase C gamma; PI3K, phosphatidylinositol 3-kinases.

**Figure 3 cells-11-03143-f003:**
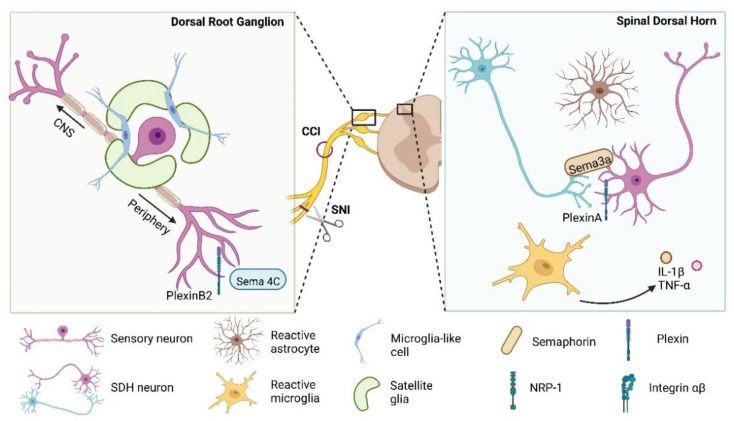
**Semaphorin–plexin signaling in mice model of pain.** Schematic representation of known semaphorins and their receptor expressed in dorsal root ganglia and the spinal dorsal horn after nerve injury in pain models of mice. CCI, chronic constriction injury; SNI, spared nerve injury; NRP-1, Neuropilin-1; Sema, Semaphorin; IL-1β, Interleukin-1β; TNF-α, Tumor necrosis factor alpha.

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
