# Peer review of "Axon Guidance Molecules and Pain"

_cells, 2022, doi:10.3390/cells11193143_

Round 1
Reviewer 1 Report
This article by Samo and Simonetti summarizes emerging evidence about the putative function of three important families of ligand-receptors and their triggered pathways in pain. The manuscript is easy to read and it is pertinent because no previous reviews have made a compilation of the current knowledge on this regard.
However, there are a multiple issues that need to be addressed:
Major points:
1- Fig1. (A). It is not clear for a non-expert view which are the two pain models that they authors refer to. Each model should be represented in a different mouse or alternatively each model indicated in a different color and they should be briefly described in the figure legend. (B). The word “induction” should be added after “…neuropathic pain…” in the figure legend. (C). Why do they represent the oligos? Not much is said about them, so they should be removed from the cartoon.
2- Fig2. The high mag synapses should be colored with the same colors than the low mag neurons.
3- In the introduction, the paragraphs dedicated to describe the Ephs, Wnt and Semas for the first time should be shorter and they should be better placed at the beginning of the sections dedicated to each family.
4- Rather than “Axon Guidance Molecules in Pain” I believe the title should be “Axon Guidance Molecules and Pain” because many of the experimental data mentioned in the manuscript do not really distinguish whether these proteins are directly involved in pain or in other associated processes.
5- The section dedicated to the ncRNA should be eliminated. Instead, the paragraphs explaining the effects of the different ncRNAs should be integrated in the appropriated sections depending of the target of each ncRNA. The notion that these pathways are also interconnected at the ncRNA levels can be mentioned in the conclusion section.
Minor points and misspellings
1- Pg2. Line 54. Add RTK after receptor tyrosine kinases.
2- Pg3. Line 116. The planar cell polarity pathway is commonly known as PCP, not PLP.
3- Pg3. Line 118: “resulting in Wnt-dependent transcription” should be changed to “resulting in betacatenin-dependent transcription”.
4- Pg3. Line 126-128. The sentence is redundant with previous sentences.
5- Pg3. Lines 130-132. That sentence is reiterative and totally dispensable.
6- Pg3. Line 133. “Indeed”. May be eliminated.
7- Pg3. Line 134. Patient should be in plural, add an “s”.
8- Pg3. Line 135. The references that demonstrate the involvement of the Wnt signaling in the context of pain in the last decade should be added at the end of the sentence.
9- Pg3. Line 138. The sentence “Preclinical pain models…..level of the spinal cord” should be eliminated.
10- Pg5. Line 175-177. The entire sentence should be eliminated.
11- Pg5. Line 184. “Interestingly, He et al, shows that Wnt 5b….”, should be substituted by “More recently, He et al. have shown that Wnt5b…”
12- Pg5. Line 206. …Wnt signaling can modulate pain sensation in different ways:… “ should be substituted by: “Wnt signaling can modulate pain sensation by acting directly on neurons and regulating synaptic plasticity or by recruiting non-neuronal cells such as microglia and astrocytes”.
13- Pg5. Line 215. Here the authors make reference to the activation of the Wnt/betacat signaling. However, in differentiated neurons accumulation of betacatenin has been associated to both the canonical Wnt pathway and alternatives Wnt pathways that do not involve activation of Betacatenin-transcription (https://doi.org/10.1146/annurev.neuro.31.060407.125649), Morenilla-Palao et al., 2020) Therefore, they should specify what of these pathways do they refer to or rephrase the sentence in a different manner that reflexes this.
14- Pg6. Line 231. “…synaptic plasticity in spinal cord”. The article “the” should be added: “…synaptic plasticity in the spinal cord”.
15- Pg6. Line 250. “…of several neurodegenerative diseases such as chronic pain” should be rephrased because chronic pain is not a neurodegenerative disease.
16- Pg6. Line 252. “…modulates the function of neurons and the plasticity of the neural circuits” would sound better: “modulates neuronal function and the plasticity of neural circuits”.
17- Pg6. Line 266. “Wnt pathways modulates cell proliferation. Remove “s” in modulates.
18- Pg9. Line 392. The complete name of GPI needs to be stated: Glicosifosfatidilinositol (GPI).
19- Pg9. Line 426. A Ref for the statement that ADAM10 can cut ephrins at the membrane level is needed.
20- Pg12. Line 547-48. This sentence does not make sense. Something is missing.
21- Pg12. Line 563-65. This sentence is redundant with the sentence in line 629-30. Remove it from here.
22- Pg13. Line 574. “Dependently” should be replaced by “depending”.
23- Pg13. Line 588. “…in the different pain conditions, whether there is a diverse …” should be: “ …in the different pain conditions or, whether there is a diverse…”.
24- Pg14. Line 632. “Big” family should be replaced by: “large” family.
25- Pg15. Line 681-83. The two first sentences: In the nervous system…….well stablished.” Should be eliminated. The paragraph should start as follows: Sema-Plnx signaling was originally discovered as repulsive axon guidance molecules (). However, over the past years,….”
26- Pg15. Line 695. “..new findings point out the role of Semaphorin …should be: “…new findings point out a role for Semaphorin…
27- Pg15. Line 697. “Given their role in regulating synaptic plasticity and” should be eliminated and the sentence should start by: “Because synaptic plasticity…..”
28- Pg15. Line 701: “Nevertheless, evidence has emerged in the last decade on the involvement of these guidance moleules in pain” should be changed by “…but evidence have been recently reported”.
29- Pg15. Line 704. The first sentence of this paragraph should be eliminated.
30- Pg15. Line 705. “During the development of nervous system” is missing the article: “During the development of the nervous sytem…”
31- Pg15. Line 720. “…of myelinated nerve terminals (233). Therefore, these studies pointed out anti-nociceptive effect of Sema3A” should be replaced by: “of myelinated nerve terminals (233) suggesting an anti-nociceptive effect of Sema3A”.
32- Pg18. Line 831. “Interestingly, this duality is also maintained…”. “This” should be replaced by “a”.
33- Pg19. Line 884: “Wnt, semaphorins and Ephrins are the best known axonal guidance molecules” should be replaced by “Wnt, semaphorins and Ephrins are well-known axonal guidance molecules”.
Reviewer 2 Report
This is a well-organized, well-written and sound review, addressing molecular mechanisms which are likely to participate in the emergence of chronic pain. The summary figures are clear and helpful.
Comment:
The Conclusions are not very informative and do not indicate how to move forward from a translational perspective as well as in relation to the interaction between axon guidance molecules, and other molecular aspects, e.g. neuroinflammatory mediators. This section of the review should be more specific and future-oriented.
Reviewer 3 Report
This is an interesting review paper on the role of axon guidance molecules such as Wnt, ephrins & semaphorins in pain. The paper is clear and well organized. I have only a few relatively minor remarks. In particular, please define all the terms used in this review.
1. Page 1, line 29, "...and becomes chronic."
2. Page 1, line 35, the end of the sentence is not clear. Please revise.
3. Line 37, please explain the origin of the term Wnt.
4. Page 2, line 50, activity-dependent.
5. Page 5, line 211, long-term potentiation (LTP).
6. Lines 357-359, the text is in grey. Please correct. Same remark for line 853.
7. The introduction section should contain a paragraph on the currently key molecules identified in pain before starting to develop the paper on axon guidance molecules.
